# Graph Self-Supervised Learning with Learnable Structural and Positional Encodings

## Abstract

We propose a novel framework that addresses a critical limitation in Graph Self-Supervised Learning (GSSL) for graph classification: the underestimation of topological information. Traditional GSSL, despite its success in various benchmarks, often fails to fully leverage the expressive power of Graph Neural Networks (GNNs), particularly in capturing complex structural properties. This limitation stems from two main factors: (1) the inadequacy of conventional GNNs in representing sophisticated topological features, and (2) the focus of self-supervised learning solely on final graph representations. To address these issues, we introduce *GenHopNet*, a GNN framework that integrates a k-hop message-passing scheme, enhancing its ability to capture local structural information without explicit substructure extraction. We theoretically demonstrate that *GenHopNet* surpasses the expressiveness of the classical Weisfeiler-Lehman (WL) test for graph isomorphism. Furthermore, we propose a structural- and positional-aware GSSL framework that incorporates topological information throughout the learning process. This approach enables the learning of representations that are both sensitive to graph topology and invariant to specific structural and feature augmentations. Comprehensive experiments on graph classification datasets, including those designed to test structural sensitivity, show that our methods consistently outperform most of the existing approaches in accuracy while maintaining computational efficiency. Our work significantly advances GSSL's capability in distinguishing graphs with similar local structures but different global topologies.

## CCS Concepts

• **Computing methodologies → Machine learning**; • **Machine learning approaches → Neural networks**.

## Keywords

Graph Self-Supervised Learning, Graph Neural network, Expressive Power of GNNs, Graph Classification, Graph Regression

**ACM Reference Format:**
Anonymous Author(s). 2025. Graph Self-Supervised Learning with Learnable Structural and Positional Encodings. In *Proceedings of International World Wide Web Conference (WWW '25)*. ACM, TBD, TBD, TBD, 12 pages. https://doi.org/XXXXXXX.XXXXXXX

## 1 Introduction

In recent years, Graph Neural Networks (GNNs) have emerged as a powerful framework for analyzing graph-structured data, advancing capabilities in various tasks [26, 29, 52, 58, 62, 68, 73]. While most GNNs focus on semi-supervised learning, growing in popularity is Self-Supervised Learning (SSL) that learns graph representations without human annotations.

Graph Self-Supervised Learning (GSSL) has demonstrated comparable performance to supervised methods in various representation learning tasks, applicable to both node-level and graph-level downstream tasks [61, 76]. In this paper, we focus on graph classification, a crucial graph-level task with significant applications in areas such as molecular property prediction, social network analysis, and protein function classification [17, 20, 62]. Graph classification presents unique challenges compared to node-level tasks. It requires capturing and differentiating global structural information across different graphs, not just local neighborhoods. Graphs can vary significantly in size and structure, demanding more flexible and expressive models. To generate effective graph-level representations, models must aggregate information from all nodes and edges while preserving discriminative structural features.

Despite GSSL's success, they often fall to fully leverage the expressive power of GNNs, by not utilizing both topological and positional information for graph classification. Topological information captures the local structural relationships within the graph through the k-hop neighborhood substructure patterns (*e.g.*, triangles, cycles), while positional information, derived from Laplacian eigenvectors or random-walk diffusion, reflects the nodes' relative positions within the graph's global structure. The lack of topological and positional focus prevents GSSL distinguishing between graphs with similar local structures but different global topologies. Specifically, in graphs where nodes may have identical local structures (*e.g.*, isomorphic or symmetrical nodes), relying only on neighboring features is inadequate for differentiation. Positional information is critical for enabling GNNs to distinguish such nodes, even when their connectivity patterns are similar. Isomorphic nodes, which cannot be differentiated based solely on their structural information, present a particular challenge. By incorporating positional encodings, GNNs can leverage this additional context to break symmetry, facilitating better differentiation among nodes. This enhancement improves the model's ability to recognize unique identities, leading to more accurate predictions in graph-related tasks.

The limitations of current GSSL methods can be attributed to two main factors: GNN Architecture Limitations and Self-Supervised Learning Constraints. Conventional GNNs typically aggregate information from immediate neighborhoods, often missing crucial structural differences that exist beyond local structures. For instance, GIN [62] has shown that certain GNN-based methods [29, 52] are less effective at distinguishing graph structures compared to Weisfeiler-Lehman (WL) based methods. Furthermore, current

GSSL methods [21, 48, 53, 67] often fail to fully leverage the complementary nature of structural and positional information, which hinders their ability to differentiate non-isomorphic graphs with similar local attributes but different global topologies.

Building on these insights, we develop a framework that fundamentally reimagines graph representation learning by innovating both GNN architecture and the self-supervised learning process. Our goal is to significantly enhance the expressiveness and representational capacity of GSSL in distinguishing non-isomorphic graphs with similar local structures but different global topologies.

To this end, we focus on two main components:

*GenHopNet* GNN: A novel GNN architecture designed to capture complex structural information beyond immediate neighborhoods. It implements a k-hop message-passing scheme that expands the receptive field of each node, allowing the model to capture long-range dependencies and global structural information.

Structural and Positional Aware Self-Supervised Learning: A new self-supervised learning framework that preserves and uses crucial topological information by incorporating both structural and positional information into learning. It overcomes limitations of methods that focus solely on final graph representations.

**Contributions.** Below we summarize our main contributions:

i. We introduce *GenHopNet*, a GNN framework that implements a k-hop message-passing aggregation scheme and surpasses the expressiveness of the WL test.

ii. We propose a structural- and positional-aware GSSL framework, namely *StructPosGSSL*, for GNN pre-training, enabling the learning of representations invariant to specific structural and feature augmentations while preserving topological and positional information.

iii. With extensive experiments on both real-world and synthetic datasets we demonstrate that our *StructPosGSSL* achieves superior performance on most graph classification benchmarks.

## 2 Related Work

GNNs are a class of neural networks designed to effectively process and represent graph-structured data. Since the development of the previous GNN models, various adaptations have emerged, including GCN [29], GAT [52], GraphSAGE [19], and GIN [62], among others. These models aim to learn distinguishing representations of graphs based on their data labels. However, annotating graph data, such as identifying categories of biochemical molecules, often requires specialized expertise, making it challenging to obtain large-scale labeled graph datasets [67]. This challenge highlights a key limitation of supervised graph representation learning.

Contrastive Learning (CL) stands out as a highly effective self-supervised technique embedding unlabeled data [31]. By bringing similar examples closer together and pushing dissimilar ones apart, CL methods—including SimCLR [10], MoCo [22], BYOL [18], MetAug [32], and Barlow Twins [69]—have demonstrated remarkable success in the realm of computer vision [44, 59].

**Graph Self-Supervised Learning (GSSL)** is a promising technique for learning representations of graph-structured data without requiring labeled examples, making it especially effective for graph classification tasks. To date, many GSSLs with unique strategies

have been proposed to enhance graph classification. These methods build on the strengths of GNNs and CL techniques [21, 49, 53].

A key focus of GSSL is the development of effective graph augmentation strategies. For instance, GraphCL [67] introduces perturbation invariance and proposes various graph augmentations, such as node dropping, edge perturbation, attribute masking, and subgraph extraction. Recognizing the limitations of using complete graphs, Subg-Con [25] advocates for subgraph sampling as a more effective method for capturing structural information. To improve the semantic depth of sampled subgraphs, MICRO-Graph [72] proposes generating informative subgraphs by learning graph motifs. Furthermore, the process of selecting suitable graph augmentations can be time-consuming and labor-intensive; JOAO [66] addresses this by introducing a bi-level optimization framework that automates the selection of data augmentations tailored to specific graph data. RGCL [33] argues that random destruction of graph properties during augmentation can lead to a loss of critical semantic information and proposes a rationale-aware approach for graph augmentation. Additionally, SPAN [34] introduces a spectral perspective for guiding topology augmentation, noting that previous work has largely concentrated on spatial domain augmentation. To address the neglect of hierarchical structures in existing GSSL methods, HGCL [27] proposes Hierarchical GSSL, which integrates node-level CL, graph-level CL, and mutual CL components. Another important aspect of GSSL is the process of negative sampling; BGRL [50] simplifies this process by eliminating the need for constructing negative samples, allowing it to scale efficiently to large graphs. To mitigate sampling bias, PGCL [35] introduces a negative sampling strategy based on semantic clustering. In contrast to existing GSSL methods, our approach enhances both global and local structural understanding. It ensures that global graph representations effectively capture complex topological similarities and differences, while local node embeddings are refined to preserve detailed structural and positional nuances. By incorporating structural and positional awareness through invariance, variance, and covariance across node features, our method improves the ability to distinguish between isomorphic and non-isomorphic graphs. This ensures that both global graph structure and local node characteristics are robustly represented and aligned.

**Enhancing GNN Expressiveness.** A substantial amount of effort has been devoted to enhancing the expressive power of GNNs beyond the 1-WL[1]. This pursuit arises from the need to capture more intricate graph structures and relationships to address complex real-world problems effectively. Broadly, there are four primary directions which GNNs can extend beyond the 1-WL level: (1) A number of studies have introduced higher-order variants of GNNs, demonstrating comparable expressiveness to k-WL with $k \geq 3$ [2]. As an example, k-order graph networks, introduced by [40], offer expressiveness that is similar to a set-based variation of k-WL. [37] introduced a 2-order graph network that maintains expressive power similar to 3-WL. Furthermore, [39] introduced a localized variant of k-WL, focusing solely on a subset of vertices within a neighborhood. Nevertheless, using these expressive GNNs presents challenges due to their intrinsic computational demands and intricate architecture. In addition, some studies aimed to integrate

---

[1]WL stands for the Weisfeiler Leman graph isomorphism test.

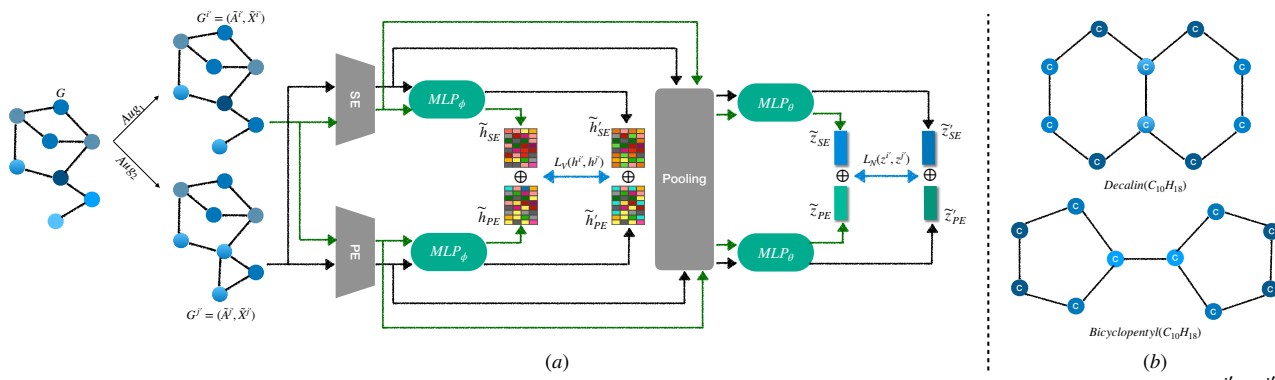

(a)                                                                              (b)

**Figure 1:** (*a*) A high-level overview of the GSSL model architecture of *StructPosGSSL* ($G$ is an input graph and $G^{i'}, G^{j'}$ are two augmented views). Our design comprises three main components: (i) a structural encoder (SE) that generates structural embeddings ($h_{SE}$ and $h'_{SE}$) for nodes based on their local structural properties; (ii) a positional encoder (PE) that generates positional embeddings ($h_{PE}$ and $h'_{PE}$) for nodes; and (iii) a pooling layer that aggregates the node representations to generate the final graph representation. Moreover, $MLP_\phi$ and $MLP_\theta$ are two shared projection heads for node representations and graph representations, respectively. (*b*) The real-world graph structures of two molecules, Decalin and Bicyclopentyl. While standard Graph SSL frameworks cannot distinguish between these molecular structures, our model successfully differentiates them.

inductive biases on isomorphism counting w.r.t predefined topological attributes such as triangles, cliques, and cycles [7, 36, 38]. These efforts similar to the traditional graph kernels, as outlined by [63]. However, the task of predefining topological characteristics needs specialised knowledge in the respective domain, a resource that is frequently not easily accessible. (3) In a different vein, there has been a recent surge in studies exploring into the notion of enhancing GNNs through the augmenting of node identifiers or stochastic features. For example, [54] introduced an approach that preserves a node's local context through the manipulation of node identifiers in a permutation-equivariant fashion. [65] developed ID-GNNs, incorporating vertex identity information in their design. [11] and [41] assigned one-hot identifiers to nodes, drawing inspiration from the principles of relational pooling. In a similar vein, [46] enriched the representational capability of GNNs by incorporating a random feature for each node. There are some other approaches modify the MPNN framework or incorporate additional heuristics to enhance their expressiveness [6, 8, 57]. (4) Some works inject positional encoding (PE) as initial node features because nodes in a graph lack inherent positional information. Canonical index PE can be assigned to the nodes in a graph. However, the model must be trained on all possible index permutations, or sampling must be employed [41]. Another direction for PE in graphs is using Laplacian Eigenvectors [14, 15], as they establish a meaningful local coordinate system while maintaining the global structure of the graph. [16] proposed a PE scheme (RWPE) based on random-walk diffusion to initialize the positional representations of nodes. These positional encoding methods such as Laplacian positional encoding [14] or RWPE [16] have a significant limitation in that they usually fail to quantify the structural similarity between nodes and their surrounding neighborhoods. Nonetheless, while these techniques have demonstrated their expressivity to go beyond 1-WL. However, it remains uncertain what further attributes they can encompass beyond the scope of 1-WL.

Despite these limitations, our method offers notable advantages. *GenHopNet* enjoys greater expressive power than the 1-WL test, providing improved node and graph-level distinction by accounting for both local and global graph structures through closed walk counts and positional information. Additionally, by incorporating edge centrality measures to enrich message-passing, *StructPosGSSL* enhances the model's ability to differentiate various types of connections, making it strictly more expressive than Subgraph MPNNs [12, 65, 71] in distinguishing certain non-isomorphic graphs.

## 3 An Expressive and Generalizable k-hop Message Passing Framework

In this section, we introduce the Expressive and Generalizable Message-Passing (EGMP) framework, designed to incorporate learnable local structural information through an aggregation method that leverages the k-hop neighborhood without the need for explicit extraction of local substructure patterns. We provide a theoretical analysis demonstrating how k-hop GNNs within this framework can achieve greater expressiveness than 1-WL.

Let $G = (V, E, \mathbf{A})$ be an undirected graph with a set of nodes $V$ and a set of edge $E$, where $|V| = m$, $|E| = e$, and $\mathbf{A} \in \mathbb{R}^{m \times m}$ is the adjacency matrix. Nodes are associated with a feature matrix $\mathbf{X} \in \mathbb{R}^{m \times z}$ with $z$ features for each node. Let $\mathbf{L} = \mathbf{D} - \mathbf{A}$ be a Laplacian matrix, where a diagonal matrix $\mathbf{D} \in \mathbb{R}^{m \times m}$, and $\mathbf{D}_{ii} = \sum_j \mathbf{A}_{ij}$. $\mathbf{L}$ be a real symmetric matrix and diagonalizable as $\mathbf{L} = \mathbf{U}\Lambda\mathbf{U}^H$. Here, $\mathbf{U} = \{u_i\}_{i=1}^m \in \mathbb{R}^m$ are orthogonal eigenvectors, $\Lambda = diag([\lambda_1, \ldots, \lambda_m]) \in \mathbb{R}^{m \times m}$ are real eigenvalues, and $\mathbf{U}^H$ is a hermitian transpose of $\mathbf{U}$.

Let $\{\!\{\cdot\}\!\}$ represent a multiset. Let $\tilde{\mathbf{A}}^k = (\tilde{\mathbf{A}}_{vu}^k)_{v,u \in V}$ where $\tilde{\mathbf{A}}_{vu}^k = \frac{\mathbf{A}_{vu}^k}{\sum_{u \in N(v) \setminus v} \mathbf{A}_{vu}^k}$ refers to a normalized value of $\mathbf{A}_{vu}^k$, and $\mathbf{X} \in \mathbb{R}^{m \times z}$ be the matrix of input feature vectors with each $\mathbf{x}_v \in \mathbb{R}^z$ corresponding to each vertex $v \in V$. We indicate the feature vector of vertex $v$ at the $t^{\text{th}}$ layer as $\mathbf{h}_v^{(t)}$ and set $\mathbf{h}_v^{(0)} = \mathbf{x}_v$. Then, the definition of the

(t+1)$^{\text{th}}$ layer in an aggregation scheme is given as:

$$\mathbf{M}_e^{(t)} = \text{AGG}^E\left(\{\!\{(\mathbf{A}_{vu}, \mathbf{h}_u^{(t)}, e_{uv}^b, e_{uv}^c)|u \in \mathcal{N}(v)\}\!\}\right), \quad (1)$$

$$\mathbf{M}_u^{(t)} = \text{AGG}^{N^k}\left(\{\!\{(\tilde{\mathbf{A}}_{vu}^k, \mathbf{h}_u^{(t)})|u \in \mathcal{N}^k(v)\}\!\}\right); k \geq 2, \quad (2)$$

$$\mathbf{M}_v^{(t)} = \text{AGG}^I\left(\{\!\{\mathbf{A}_{vv}^k, \mathbf{h}_v^{(t)}\}\!\}\right); k \geq 2, \quad (3)$$

$$\mathbf{h}_v^{(t+1)} = \text{COMBINE}\left(\mathbf{h}_v^{(t)}, \mathbf{M}_e^{(t)}, \mathbf{M}_u^{(t)}, \mathbf{M}_v^{(t)}\right). \quad (4)$$

The above equations define a process for aggregating messages in our GNN, where:

i. $\text{AGG}^E(\cdot)$ in Eq. 1 computes the edge-level aggregated message $M_e^{(t)}$ for vertex $v$ based on the node features and edge attributes of its neighbors $u \in \mathcal{N}(v)$ within the 1-hop neighborhood;

ii. $\text{AGG}^{N^k}(\cdot)$ in Eq. 2 aggregates a normalized message $M_u^{(t)}$ from the $k$-hop neighbors ($k \geq 2$) of vertex $v$ weighing their contributions by the normalized adjacency matrix $\tilde{\mathbf{A}}_{vu}^k$;

iii. $\text{AGG}^I$ in Eq. 3 calculates the self-message $\mathbf{M}_v^{(t)}$ for vertex $v$ using its own adjacency information and features, focusing on capturing closed-walks of length up to $k$ ($k \geq 2$) that return to node $v$ itself;

iv. Eq. 4 combines the current feature vector $\mathbf{h}_v^{(t)}$ of vertex $v$ with the aggregated messages $\mathbf{M}_e^{(t)}$, $\mathbf{M}_u^{(t)}$, and $\mathbf{M}_v^{(t)}$ to update final node representation of $v$ for the next layer.

We use the above set of equations to compute the graph's topological information. For the positional information, we only use Eq. 1 by replacing node features $\mathbf{h}_u^{(t)}$ with positional features $\mathbf{h}_{u,pos}^{(t)}$.

In more detail, $\mathbf{M}_u^{(t)}$ is a message aggregated from neighbors of node $v$, using their normalized coefficients $\tilde{\mathbf{A}}_{vu}^k$, while $\mathbf{M}_v^{(t)}$ is the adjusted message from node $v$ to itself, considering walk lengths up to k-hop. The diagonal elements of $\mathbf{A}^k$, namely $\mathbf{A}_{vv}^k$, count the number of closed walks of length $k$ that start and end at the same node $v$. Such a mechanism highlights the importance of node $v$ within its local topology and its role in the connectivity of the graph over multiple hops. This should exhibit the following properties:

1. **Closed Walks and Connectivity:** The power $\mathbf{A}^k$ enumerates all possible walks of length $k$ in the graph. By examining $\mathbf{A}_{vv}^k$, one can infer how *connected* or *central* a node is with respect to walks of length $k$. $\mathbf{Tr}(\mathbf{A}^k)$ counts the total number of closed walks of length $k$ starting and ending at the same vertex. This measure gives a quantitative sense of the graph's connectivity:
   - **Local Connectivity:** High numbers of shorter closed walks (smaller $k$) indicate strong local connectivity. This is useful for understanding how tightly knit individual neighborhoods are within the graph.
   - **Global Connectivity:** As $k$ increases, the nature of the closed walks provides insights into the global connectivity and the presence of cycles within the graph. Any cycle in a graph is a closed walk; however, not all closed walks are cycles as cycles have the additional constraint of not repeating vertices or edges except the starting/ending vertex.

2. **Isomorphic Invariant:** This simple trick facilitates the creation of unique node representations by ensuring that each

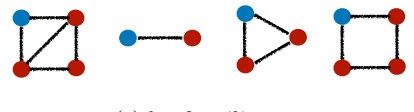

$G$    (a) $k = 2$    (b) $k = 3$    (c) $k = 4$

**Figure 2: A high-level overview of different closed-walks $k = 2, \ldots, 4$, where the blue node represents the source node. For $k = 2$, the walk can traverse between nodes multiple times, forming a non-cyclic path. For $k = 3$ and $k = 4$, the walks can capture closed cycles of length 3 and 4, respectively, effectively identifying cyclic structures in the graph.**

node possesses a distinct k-hop topological neighborhood, provided that k is sufficiently large. The sum $\sum_k \mathbf{A}_{vv}^k$ across different powers $k$ reflects the number of closed walks of varying lengths starting and ending at node $v$. For instance, the equality $\sum_k \mathbf{A}_{vv}^k = \sum_k \mathbf{A}_{v'v'}^k$, holds if nodes $v$ and $v'$ are k-hop isomorphic, implying that they share identical local connectivity patterns up to k-hops. This characteristic serves as a powerful tool for identifying and distinguishing nodes based on their structural roles within the network.

Understanding cycle and closed-walk isomorphisms is essential for elucidating the structural roles of nodes within a graph, revealing both local and global connectivity patterns. Cycle isomorphism highlights nodes that engage in similar closed-loop interactions, while closed-walk isomorphism provides insights into broader connectivity by capturing indirect relationships that contribute to the overall network topology. The definitions of cycle and closed-walk isomorphisms formalize these concepts, emphasizing their significance in analyzing graph structures.

DEFINITION 1. *Cycle Isomorphism ($\simeq_{Cycle}$): Two nodes $v$ and $v'$ are cycle isomorphic if the sum of the powers of $\mathbf{A}$ over $k$ (i.e., $\sum_k \mathbf{A}_{vv}^k$ and $\sum_k \mathbf{A}_{v'v'}^k$) considers only closed walks that are also cycles (i.e., walks that do not repeat any vertices or edges except the starting/ending vertex).*

DEFINITION 2. *Closed Walk Isomorphism ($\simeq_{ClosedWalk}$): Two nodes $v$ and $v'$ are closed walk isomorphic if the sum of the powers of $\mathbf{A}$ over $k$ (i.e., $\sum_k \mathbf{A}_{vv}^k$ and $\sum_k \mathbf{A}_{v'v'}^k$) considers all possible closed walks (i.e., walks that start and end at the same vertex, but may repeat vertices and edges).*

THEOREM 1. *The following statement is true: (a) If $\sum_k \mathbf{A}_{vv}^k \simeq_{Cycle} \sum_k \mathbf{A}_{v'v'}^k$, then $\sum_k \mathbf{A}_{vv}^k \simeq_{ClosedWalk} \sum_k \mathbf{A}_{v'v'}^k$; but not vice versa.*

Going back ot Eq. 1, $\mathbf{M}_e^{(t)}$ is a message aggregation function that considers the 1-hop neighborhood, including their edge connections and edge attributes. We enrich the 1-hop neighborhood aggregation by injecting three different centrality based edge attributes $\mathbf{e}_{uv}^c$, between given pair of nodes: (1) Edge Betweenness (EB), (2) Edge Closeness (EC), and (3) Edge Clustering Coefficients (ECC) as additional features [56]. These attributes are designed to augment the distinguishing capabilities of the model, enabling it to better differentiate between various types of connections and facilitating nuanced understanding and processing of edge-related information in graphs. This should exhibit the following properties:

1. **Local Connectivity:** Both EC and ECC provide insights into the local structure around an edge, highlighting how embedded an edge is within its immediate neighborhood and how it contributes to local connectivity and cohesiveness.
2. **Global Connectivity:** EB extends to more global properties, reflecting the strategic importance of an edge across the entire network. It helps in understanding the potential vulnerabilities of the network, identifying crucial links whose removal or failure might significantly disrupt network connectivity.
3. **Isomorphic Invariant:** All three measures (EB, EC, and ECC) are isomorphic invariant. This is because they are fundamentally based on the relationships and distances between nodes, which are preserved under graph isomorphism.

Finally, the feature vector for the next layer $\mathbf{h}_v^{(t+1)}$, is derived by combining $\mathbf{h}_v^{(t)}$, $\mathbf{M}_e^{(t)}$, $\mathbf{M}_u^{(t)}$, and $\mathbf{M}_v^{(t)}$ using the Combine function.

## 4 Generalizable k-Hop Network

Below, we present the GNN model design based on the EGMP framework that we proposed in the previous section. There are numerous ways to designing $\phi$ functions, which result in GNNs with varying levels of expressiveness. To illustrate this, we introduce a new GNN model called *GenHopNet* (Generalizable k-Hop Network), which utilizes an aggregation scheme based on our generalized message-passing framework. We demonstrate that the expressive power of GenHopNet exceeds those of the 1-WL. For each vertex $v \in V$, the feature vector for the $(t + 1)$-th layer is produced by:

$$\mathbf{h}_v^{(t+1)} = \text{Mlp}_\theta\Big[(1+\epsilon)\mathbf{h}_v^{(t)} + \sum_{u \in N(v)} \mathbf{A}_{vu}\big(\mathbf{h}_u^{(t)} + \mathbf{e}_{vu}^b + \mathbf{e}_{vu}^c\big)$$
$$+ \sum_{k=2}^{K}\big(\mathbf{A}_{vv}^k \mathbf{h}_v^{(t)} + \sum_{u \in N^k(v)} \tilde{\mathbf{A}}_{vu}^k \mathbf{h}_u^{(t)}\big)\Big]. \quad (5)$$

The $\epsilon$ is a learnable scalar param. $\mathbf{M}_v^{(t)} = \sum_{k=2}^{K} \mathbf{A}_{vv}^k \mathbf{h}_v^{(t)}$, $\mathbf{M}_u^{(t)} = \sum_{k=2}^{K} \sum_{u \in N^k(v)} \tilde{\mathbf{A}}_{vu}^k \mathbf{h}_u^{(t)}$, and $\mathbf{M}_e^{(t)} = \sum_{u \in N(v)} \mathbf{A}_{vu}(\mathbf{h}_u^{(t)} + \mathbf{e}_{vu}^b + \mathbf{e}_{vu}^c)$.
**Expressiveness analysis.** We first generalise the result of universal functions over *multisets* [62] to universal functions over *pairs of multisets* since Eq. 5 involves not only node features but also edge features $\mathbf{e}_{ij}^b$, centrality based edge features $\mathbf{e}_{ij}^c$, normalized k-hop neighboring coefficients $\tilde{\mathbf{A}}_{ij}^k$ and k-hop closed-walk coefficients $\mathbf{A}_{jj}^k$. Let $\mathcal{H}$, $\mathcal{A}$, $\mathcal{A}^k$, $\tilde{\mathcal{A}}^k$, $\mathcal{W}_1$, $\mathcal{W}_2$, and $\mathcal{W}_3$ be countable sets where $\mathcal{H}$ is a node feature space, $\mathcal{A}$ is a 1-hop neighborhood coefficient space, $\mathcal{A}^k$ is a k-hop closed-walk coefficient space, $\tilde{\mathcal{A}}^k$ is a normalized k-hop neighborhood coefficient space. Moreover, $\mathcal{W}_1 = \{\mathbf{A}_{ij}(\mathbf{h}_i + \mathbf{e}_{ij}^b + \mathbf{e}_{ij}^c)|\mathbf{A}_{ij} \in \mathcal{A}, \mathbf{h}_i \in \mathcal{H}, \mathbf{e}_{ij}^b \in \mathcal{E}^b, \mathbf{e}_{ij}^c \in \mathcal{E}^c\}$, $\mathcal{W}_2 = \{\tilde{\mathbf{A}}_{ij}^k \mathbf{h}_i|\tilde{\mathbf{A}}_{ij}^k \in \tilde{\mathcal{A}}^k, \mathbf{h}_i \in \mathcal{H}\}$, and $\mathcal{W}_3 = \{\mathbf{A}_{jj}^k \mathbf{h}_j|\mathbf{A}_{jj}^k \in \mathcal{A}^k, \mathbf{h}_j \in \mathcal{H}\}$.
The following theorem asserts that a GNN can surpass the expressiveness of 1-WL provided that our framework is sufficiently robust to differentiate structures beyond neighborhood subtrees, and the neighborhood aggregation function, $\phi$ is injective, given a sufficient number of hops where $k > 1$.

**Theorem 2.** *Let $S$ represent a GNN with an aggregation scheme $\Phi$ delineated by Eq. 1-Eq. 4. $S$ exceeds the expressiveness of 1-WL in*

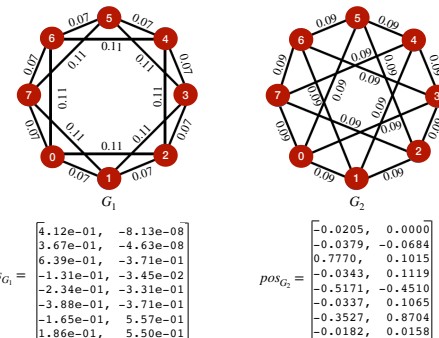

$$pos_{G_1} = \begin{bmatrix} 4.12\text{e-}01, & -8.13\text{e-}08 \\ 3.67\text{e-}01, & -4.63\text{e-}08 \\ 6.39\text{e-}01, & -3.71\text{e-}01 \\ -1.31\text{e-}01, & -3.45\text{e-}02 \\ -2.34\text{e-}01, & -3.31\text{e-}01 \\ -3.88\text{e-}01, & -3.71\text{e-}01 \\ -1.65\text{e-}01, & 5.57\text{e-}01 \\ 1.86\text{e-}01, & 5.50\text{e-}01 \end{bmatrix} \quad pos_{G_2} = \begin{bmatrix} -0.0205, & 0.0000 \\ -0.0379, & -0.0684 \\ 0.7770, & 0.1015 \\ -0.0343, & 0.1119 \\ -0.5171, & -0.4510 \\ -0.0337, & 0.1065 \\ -0.3527, & 0.8704 \\ -0.0182, & 0.0158 \end{bmatrix}$$

**Figure 3: A pair of non-isomorphic graphs where positional encoding (with a dimension of 2) and EB attributes outperform the 1-WL test in distinguishing between graphs $G_1$ and $G_2$, allowing for the detection of structural differences that the 1-WL test fails to capture.**

*identifying non-isomorphic graphs, provided that $S$ operates over a sufficient number of hops, $k > 1$, and also meets the following criteria:*

(1) $\Phi\Big(\mathbf{h}_v^{(t)}, \{\!\{(\mathbf{A}_{vu}, \mathbf{h}_u^{(t)}, \mathbf{e}_{uv}^b, \mathbf{e}_{uv}^c)|u \in \mathcal{N}(v)\}\!\}, \{\!\{(\tilde{\mathbf{A}}_{vu}^k, \mathbf{h}_u^{(t)})|u \in \mathcal{N}^k(v)\}\!\}, \{\!\{(\mathbf{A}_{vv}^k, \mathbf{h}_v^{(t)})\}\!\}\Big)$ *is injective;*

(2) *The graph-level readout function of $S$ is injective.*

**Lemma 1.** *Given two distinct pairs of multisets $\mathbf{H}_1, \mathbf{H}_1' \in \mathcal{H}$, $\mathbf{W}_1, \mathbf{W}_1' \in \mathcal{W}_1$, $\mathbf{W}_2, \mathbf{W}_2' \in \mathcal{W}_2$, $\mathbf{W}_3, \mathbf{W}_3' \in \mathcal{W}_3$, there exists a function $f$ such that the aggregation function $\pi(\mathbf{h}_v, \mathbf{H}_1, \mathbf{W}_1, \mathbf{W}_2, \mathbf{W}_3)$ and $\pi(\mathbf{h}_v', \mathbf{H}_1', \mathbf{W}_1', \mathbf{W}_2', \mathbf{W}_3')$ defined as $\pi(\mathbf{h}_v, \mathbf{H}_1, \mathbf{W}_1, \mathbf{W}_2, \mathbf{W}_3) = \sum_{\mathbf{h} \in \mathbf{H}_1, \mathbf{w}_1 \in \mathbf{W}_1} f(\mathbf{h}, \mathbf{w}_1) + \sum_k \big(f(\mathbf{h}_v, \mathbf{w}_3) + \sum_{\mathbf{h} \in \mathbf{H}_1, \mathbf{w}_2 \in \mathbf{W}_2} f(\mathbf{h}, \mathbf{w}_2)\big)$ and $\pi(\mathbf{h}_v', \mathbf{H}_1', \mathbf{W}_1', \mathbf{W}_2', \mathbf{W}_3') = \sum_{\mathbf{h} \in \mathbf{H}_1', \mathbf{w}_1 \in \mathbf{W}_1'} f(\mathbf{h}, \mathbf{w}_1) + \sum_k \big(f(\mathbf{h}_v', \mathbf{w}_3) + \sum_{\mathbf{h} \in \mathbf{H}_1', \mathbf{w}_2 \in \mathbf{W}_2'} f(\mathbf{h}, \mathbf{w}_2)\big)$ are unique, respectively.*

**Lemma 2.** *Expanding upon Lemma 1, we introduce an extended aggregation function $\pi'(\mathbf{h}_v, \mathbf{H}, \mathbf{W}_1, \mathbf{W}_2, \mathbf{W}_3)$, which incorporates the feature vector of the central node $\mathbf{h}_v$ and the multisets $\mathbf{H} \in \mathcal{H}$, $\mathbf{W}_1 \in \mathcal{W}_1$, $\mathbf{W}_2 \in \mathcal{W}_2$, and $\mathbf{W}_3 \in \mathcal{W}_3$. There exists a function $f$ such that $\pi'(\mathbf{h}_v, \mathbf{H}, \mathbf{W}_1, \mathbf{W}_2, \mathbf{W}_3) = (1 + \epsilon)f(\mathbf{h}_v) + \sum_{\mathbf{h} \in \mathbf{H}, \mathbf{w}_1 \in \mathbf{W}_1} f(\mathbf{h}, \mathbf{w}_1) + \sum_k \big(f(\mathbf{h}_v, \mathbf{w}_3) + \sum_{\mathbf{h} \in \mathbf{H}, \mathbf{w}_2 \in \mathbf{W}_2} f(\mathbf{h}, \mathbf{w}_2)\big)$ is unique for any distinct quintuples $(\mathbf{h}_v, \mathbf{H}, \mathbf{W}_1, \mathbf{W}_2, \mathbf{W}_3)$, where $\mathbf{h}_v \in \mathcal{H}$, $\mathbf{w}_3 \in \mathcal{W}_3$, and $\epsilon$ is an arbitrary real number.*

**Corollary 1.** *GenHopNet exhibits greater expressiveness compared to 1-WL when evaluating non-isomorphic graphs.*

Our proposed graph neural network model, which operates on up to k-hop local information for feature aggregation, denoted as $S$, surpasses the expressiveness of 1-WL in distinguishing non-isomorphic graphs. This is achieved by ensuring the uniqueness of feature representations through the extended aggregation function $\pi'(\mathbf{h}_v, \mathbf{H}, \mathbf{W}_1, \mathbf{W}_2, \mathbf{W}_3)$, as established by Lemma 2. Thus, $S$ can capture and differentiate structural nuances beyond what 1-WL can achieve, making it a powerful tool for graph classification tasks.

**Complexity Analysis:** Similar to GIN [23], GenHopNet is computationally efficient, with its time and memory complexities scaling linearly in relation to the number of edges in the graph. The time

and space complexities of *GenHopNet* are $O(tezd)$ and $O(e)$, respectively, where $e$ denotes the number of edges in the graph, $t$ represents the number of layers, and $z$ and $d$ correspond to the dimensions of the input and output feature vectors.

## 5 Graph Self-Supervised Learning Framework

In this section, we introduce Structural and Positional GSSL (*StructPosGSSL*), a new class of graph self-supervised learning framework based on structural and positional information within graphs.

### 5.1 Data Augmentation for Graph

The goal of data augmentation is to produce consistent, identity-preserving positive samples of a specific graph. In this work, we use two main types of augmentation strategies: structural augmentation and feature augmentation [67]. In structural augmentation, three distinct strategies are considered: (1) Subgraph Induction by Random Walks (RWS), (2) Node Dropping (ND), and (3) Edge Dropping (ED). For feature augmentation, we employ three different approaches: (1) Feature Dropout (FD), (2) Feature Masking (FM), and (3) Edge Attribute Masking (EAM). In our work, we generate different augmented graphs from a single input graph $(\mathbf{A}, \mathbf{X})$, resulting in two correlated views, namely $(\tilde{\mathbf{A}}^{i'}, \tilde{\mathbf{X}}^{i'})$ and $(\tilde{\mathbf{A}}^{j'}, \tilde{\mathbf{X}}^{j'})$.

### 5.2 Expressive/Genralizable Graph Encoders

For each augmented view, we process it through two distinct GNN encoders. One encoder is dedicated to structural encoding via the proposed *GenHopNet*. Alongside structural features in the *GenHopNet*, we employ another encoder for capturing positional information. We initiate the positional feature vectors using Laplacian eigenvectors, as outlined in [14]. This second encoder, also a GNN, applies Eq.1 with the COMBINE function, i.e., $\mathbf{h}_{v,pos}^{(t+1)} = \text{COMBINE}\left(\mathbf{h}_{v,pos}^{(t)}, \mathbf{M}_e^{(t)}\right)$, and leverages the spectral properties of the graph Laplacian. This approach targets the smallest non-trivial eigenvalues to derive meaningful positional encodings that accurately reflect the structural roles of nodes within the graph. By focusing on the spectral characteristics of the Laplacian, this encoding strategy effectively captures both the local connectivity of nodes and the broader topology of the graph, significantly enhancing the model's capability to understand and manage complex graph structures. Each encoder outputs node representations and a final graph representation for augmented views. We then pass them through another shared projection head (an MLP) to obtain the final structural and positional representations for both nodes and graphs. Next, we concatenate the structural features with positional features for node representations and graph representations separately, ensuring a comprehensive integration of both structural and positional data. To facilitate the end-to-end training of encoders and generate comprehensive node and graph representations that are independent of specific downstream tasks, we implement a loss function that merges NT-Xent [10] and refined VICReg [3].

### 5.3 Training Pipeline

In the pursuit of effective self-supervised learning frameworks, the quality of learned representations hinges critically on the choice of loss functions. To enable end-to-end training of the encoders and to develop robust graph and node representations that are independent of downstream tasks, we employ the NT-Xent [10] loss to learn

| Method | CSL | SR25 |
|---|---|---|
| GCN | 10.0 ± 0.0 | 6.6 ± 0.0 |
| GIN | 10.0 ± 0.0 | 6.6 ± 0.0 |
| 3WLGNN | 97.8 ± 10.9 | - |
| 3-GCN | 95.7 ± 14.8 | 6.6 ± 0.0 |
| GCN-RNI | 16.0 ± 0.0 | 6.6 ± 0.0 |
| StructPosGSSL-SA | 98.6 ± 2.8 | 100.0 ± 0.0 |
| StructPosGSSL-FA | 98.3 ± 2.5 | 100.0 ± 0.0 |

**Table 1: Classification accuracy (%) on the test set for CSL and SR25 datasets.**

the graph representations and use the refined VICReg [3] loss as a regularization term to learn the node representations. The NT-Xent loss, which maximizes discriminative power between positive and negative samples of graph representations, while VICReg's regularization terms that ensure a balanced and non-redundant spread of node features across dimensions to reduce the representational collapse. This integration stabilizes training in self-supervised setup and enriches graph representations for downstream tasks.

The NT-Xent loss, is defined by the formula:

$$L_N(\mathbf{z}^{i'}, \mathbf{z}^{j'}) = -log \frac{exp(sim(\mathbf{z}^{i'}, \mathbf{z}^{j'})/\tau)}{\sum_{k=1}^{2n} \mathbf{1}_{[k \neq i']} exp(sim(\mathbf{z}^{i'}, \mathbf{z}^k)/\tau)}, \quad (6)$$

where $\mathbf{z}^{i'}, \mathbf{z}^{j'} \in \mathbb{R}^{\tilde{d}}$ are graph representations for two augmented views, with $\tilde{d}$ denoting the embedding size of each graph augmented view in the dataset. The parameter $\tau$ is the temperature scaling parameter, and $\mathbf{1}_{[k \neq i]}$ is an indicator function that is $\mathbf{1}$ if $k \neq i$ and 0 otherwise. Here $sim(\mathbf{z}^{i'}, \mathbf{z}^{j'})$ is a similarity function, typically the cosine similarity, between two vectors $\mathbf{z}^{i'}$ and $\mathbf{z}^{j'}$.

We combine the refined VICReg loss with the contrastive NT-Xent loss to create a unified loss function that maximizes mutual information between graph embeddings while preserving structural and positional node alignment. This unified approach not only aligns embeddings for isomorphic node pairs but also maintains diversity and enhances the model's topological discriminative power. By enabling the model to differentiate between nodes with subtle structural or positional differences, this method is crucial for accurately identifying both isomorphic and non-isomorphic node pairs. In this work, the refined VICReg loss is specifically adapted for correspondence node alignment to improve isomorphic graph representation learning in a self-supervised setting.

The refined VICReg loss is given by:

$$L_V(\mathbf{h}^{i'}, \mathbf{h}^{j'}) = \lambda_{inv} \cdot L_{inv}(\mathbf{h}^{i'}, \mathbf{h}^{j'}) + \lambda_{var} \cdot L_{var}(\mathbf{h}^{i'}, \mathbf{h}^{j'}) + \lambda_{cov} \cdot L_{cov}(\mathbf{h}^{i'}, \mathbf{h}^{j'}) \quad (7)$$

where $\lambda_{inv}, \lambda_{var}$ and $\lambda_{cov}$ are weighting factors for the invariance, variance, and covariance components, respectively. The Invariance Loss $L_{inv}(\mathbf{h}^{i'}, \mathbf{h}^{j'}) = \frac{1}{m} \sum_i ||\mathbf{h}_i^{i'} - \mathbf{h}_i^{j'}||_2^2$, where $\mathbf{h}^{i'}, \mathbf{h}^{j'} \in \mathbb{R}^{m \times d}$ represent the node representations of the two augmented views. The Variance Loss $L_{var}(\mathbf{h}^{i'}, \mathbf{h}^{j'}) = \frac{1}{d} \sum_{j=1}^{d} \left| max(0, \gamma - std(\mathbf{h}_j^{i'}, \epsilon)) - max(0, \gamma - std(\mathbf{h}_j^{j'}, \epsilon)) \right|$. The Covariance Loss $L_{cov}(\mathbf{h}^{i'}, \mathbf{h}^{j'}) = \frac{1}{d} \sum_{i \neq j} \left| [\mathbf{C}(\mathbf{h}^{i'})]_{i,j}^2 - [\mathbf{C}(\mathbf{h}^{j'})]_{i,j}^2 \right|$, where $\mathbf{C}(\mathbf{h}^{i'}) = \frac{1}{m-1} \sum_{i=1}^{m} (\mathbf{h}_i^{i'} - \tilde{\mathbf{h}}^{i'})(\mathbf{h}_i^{i'} - \tilde{\mathbf{h}}^{i'})^T$, with $\tilde{\mathbf{h}}^{i'} = \frac{1}{m} \sum_{i=1}^{m} \mathbf{h}_i^{i'}$ denoting the mean over feature vectors. The Invariance Loss captures the inherent structure of the graph, maintaining node representations'

structural and positional properties despite augmentation. The Variance Loss helps distinguish non-isomorphic nodes, enhancing the model's ability to capture structural differences. Lastly, the Covariance Loss ensures that node embeddings reflect diverse structural and positional aspects, improving the expressiveness and discriminative power of the representations.

The total combined loss for integrating NT-Xent and refined VICReg is simply the sum of these two losses:

$$L_T(\mathbf{z}^{i'}, \mathbf{z}^{j'}, \mathbf{h}^{i'}, \mathbf{h}^{j'}) = L_N(\mathbf{z}^{i'}, \mathbf{z}^{j'}) + \mu \cdot L_V(\mathbf{h}^{i'}, \mathbf{h}^{j'}), \quad (8)$$

where $\mu$ is the weighting factor for VICReg loss. This unified loss function effectively integrates isomorphism preservation with representation expressiveness and diversity.

THEOREM 3. StructPosGSSL[2] *is more expressive than subgraph MPNNs in distinguishing certain non-isomorphic graphs.*

## 6 Numerical Experiments

In this section, we evaluate our self-supervised learning framework on graph classification benchmark tasks. The results from our models are statistically significant with a 95% confidence level. We evaluate *StructPosGSSL* on graph classification benchmark tasks and compare their performance with leading baselines to address the following questions:

**Q1.** How effective are *StructPosGSSL* in small graph classification tasks based on empirical performance?

**Q2.** How effective are *StructPosGSSL* in large graph classification tasks based on empirical performance?

**Q3.** How effective is *StructPosGSSL* for isomorphism testing in synthetic graph classification tasks?

**Q4.** How do structural and positional encodings impact overall performance?

In this following sections, we analyze the experimental results to address the four previously mentioned questions.

### 6.1 Experiments on Small Graphs

We use eight datasets from two categories: (1) bioinformatics datasets: MUTAG, PTC-MR, NCI1, and PROTEINS [13, 30, 47, 55]; (2) social network datasets: IMDB-B, IMDB-M, COLLAB and RDT-M5K [63].

We compare our method against fourteen baseline approaches: (1) Graph kernel methods: WL subtree kernel (WL) [47], WL-OA [30], RetGK [74], P-WL [45], and WL-PM [43]; (2) GNN-based methods: PATCHY-SAN [42], DGCNN [70], CAPSGNN [60], and GIN [62]; (3) Unsupervised methods: InfoGraph [48], GraphCL [67], MVGRL [21], AutoGCL [64], and JOAO [66].

Specifically, the *GenHopNet* encoder models are initially trained in an unsupervised manner, and the resulting embeddings are then input into a linear classifier to accommodate the labeled data. Then, for fair comparison, we execute our method using ten random splits [75] and utilize the 10-fold cross-validation method and present the best mean accuracy (%) along with the standard deviation. The results are presented in table 2 and 3. We have two settings: (1) StructPosGSSL-SA, which considers structure augmentation, and (2) StructPosGSSL-FA, which considers feature augmentation. In both settings, we employ the Adam optimizer [28], with hidden

[2]The code is available at: https://anonymous.4open.science/r/StructPosGSSL-73D0

dimension of 128, weight decay is 0.0003, a 2-layer MLP with batch normalization, 100 epochs, positional encoding dimension of 6, a dropout rate of 0.5, and a temperature scaling parameter $\tau$ of 0.10. We choose a batch size from {32, 64, 128, 256} and a number of hops $k \in \{2, 3, 4, 5, 6\}$. We use $\lambda_{inv} = 1, \lambda_{var} = 3, \lambda_{cov} = 2$, and $\mu = 0.5$ for MUTAG and $\lambda_{inv} = 1, \lambda_{var} = 23, \lambda_{cov} = 23$, and $\mu = 0.0003$ for PTC-MR, and $\lambda_{inv} = 1, \lambda_{var} = 24, \lambda_{cov} = 24$, and $\mu = 0.005$ for the remaining datasets. The readout function, as described in [62], is utilized, which involves concatenating representations from all layers to derive a final graph representation.

To address **Q1**, in Tables 2 & 3, StructPosGSSL outperforms the best baseline by 0.4% (PATCHY-SAN), 1.1% (CapsGNN), 1.1% (WL-OA, MVGRL), 0.5% (WL-PM), and 0.2% (GIN) on the datasets MUTAG, PTC-MR, IMDB-B, IMDB-M, and RDTM5K, respectvely.

These gains are a reflection of the inherent characteristics of the datasets. Graphs with smaller diameters, *i.e.*, IMDB-B, IMDB-M, PTC-MR, and MUTAG, feature nodes that are closer together, promoting localized interactions that enable GNNs to capture both local and global information effectively, even with few message-passing steps. In such cases, structural encoding with closed walks is particularly beneficial, as it differentiates local structures by capturing repeated node interactions and identifying cycles. Conversely, datasets such as NCI1 and COLLAB, with larger diameters, present increased structural complexity, making it challenging to capture patterns using closed walks alone due to the difficulty of accounting for distant node interactions with limited local information.

### 6.2 Experiments on Large Graphs

We utilize five large graph datasets from the Open Graph Benchmark (OGB) [24], comprising one molecular graph dataset (ogbg-moltoxcast, ogbg-moltox21, ogbg-molhiv, ogbg-molpcba) and one protein-protein association network (ogbg-ppa). We compare our approach with the following methods that have reported results on the aforementioned OGB datasets: GIN and GIN+VN [24], GSN [9], ID-GNNs [65], Deep LRP [11], GraphSNN [57], DS-GNN (EGO+), DSS-GNN (EGO+) [5], and POLICY-LEARN [4].

For large graph datasets, we adopt the same experimental framework as outlined in [24]. Our evaluation process is divided into two distinct learning phases. In the initial phase, the models are trained in a self-supervised fashion using only node features and graph structure without any label data. Subsequently, in the second phase, the representations generated by the GNN encoders during the first phase are fixed in place and employed to train, validate, and test the models using a straightforward linear classifier.

We utilize the Adam optimizer with a learning rate of 0.001, a batch size of 32, dropout of 0.5, positional encoding dimension of 6, and run training for 100 epochs across all datasets. We use a 2-layer MLP with a hidden dimension of 200 and a temperature scaling parameter $\tau$ of 0.10 for both settings. We choose $\lambda_{inv} = 1, \lambda_{var} = 24, \lambda_{cov} = 24$, and $\mu = 0.005$ for all datasets. The classification accuracy results are presented in Table 4.

To address **Q2**, in Table 4, StructPosGSSL consistently outperforms all the baseline methods across all OGB graphs listed. StructPosGSSL surpasses best results of existing GNNs by 0.50% (POLICY-LEARN), 0.55% (DSS-GNN (EGO+)), 0.32% (GIN+VN), 0.24% (GraphSNN), and 0.42% (GIN+VN) on the datasets ogbg-moltoxcast, ogbg-moltox21, ogbg-molhiv, ogbg-ppa, and ogbg-molpcba, respectively.

| | Method | MUTAG | PTC-MR | NCI1 | PROTEINS |
|---|---|---|---|---|---|
| Graph kernel methods | WL | 90.4 ± 5.7 | 59.9 ± 4.3 | 86.0 ± 1.8 | 73.8 ± 3.9 |
| | WL-OA | 84.5 ± 1.7 | 63.6 ± 1.5 | 86.1 ± 0.2 | 74.2 ± 0.4 |
| | RetGK | 90.3 ± 1.1 | 62.5 ± 1.6 | 84.5 ± 0.2 | 72.3 ± 0.6 |
| | P-WL | 90.5 ± 1.3 | 64.0 ± 0.8 | 85.4 ± 0.1 | 70.4 ± 0.1 |
| | WL-PM | 87.7 ± 0.8 | 61.4 ± 0.8 | **86.4 ± 0.2** | 73.6 ± 0.2 |
| GNN-based methods | PATCHY-SAN | 92.6 ± 4.2 | 60.0 ± 4.8 | 78.6 ± 1.9 | 73.1 ± 2.4 |
| | DGCNN | 85.8 ± 0.0 | 58.6 ± 0.0 | 74.4 ± 0.0 | 70.0 ± 0.9 |
| | CapsGNN | 86.6 ± 1.5 | 66.0 ± 1.8 | 78.3 ± 1.3 | 73.1 ± 4.8 |
| | GIN* | 89.4 ± 5.6 | 64.6 ± 7.0 | 82.7 ± 1.7 | **75.1 ± 5.1** |
| Unsupervised methods | InfoGraph | 89.0 ± 1.1 | 61.7 ± 1.4 | 76.2 ± 1.4 | 74.4 ± 0.3 |
| | GraphCL | 86.8 ± 1.3 | 61.3 ± 2.1 | 77.9 ± 0.4 | 74.4 ± 0.4 |
| | MVGRL | 89.7 ± 1.1 | 62.5 ± 1.7 | 77.0 ± 0.8 | - |
| | AutoGCL | 88.6 ± 1.0 | - | 82.0 ± 0.3 | 75.8 ± 0.4 |
| | JOAO | 87.3 ± 1.0 | - | 78.0 ± 0.4 | 74.5 ± 0.4 |
| Our work | StructPosGSSL-SA | **93.0 ± 5.3** | **67.1 ± 4.9** | 82.9 ± 3.8 | 75.8 ± 2.6 |
| | StructPosGSSL-FA | 91.5 ± 2.5 | 66.5 ± 3.7 | 82.5 ± 3.5 | 75.3 ± 2.9 |

**Table 2: Classification accuracy (%) on bioinformatics datasets averaged over 10 runs.**

| | Method | IMDB-B | IMDB-M | COLLAB | RDTM5K |
|---|---|---|---|---|---|
| Graph kernel methods | WL | 73.8 ± 3.9 | 50.9 ± 3.8 | 78.9 ± 1.9 | 52.5 ± 2.1 |
| | WL-OA | 74.2 ± 0.4 | 51.3 ± 0.2 | 80.7 ± 0.1 | - |
| | RetGK | 72.3 ± 0.6 | 48.7 ± 0.6 | **81.0 ± 0.3** | 56.1 ± 0.5 |
| | P-WL | 70.4 ± 1.1 | 50.9 ± 0.3 | - | - |
| | WL-PM | 73.6 ± 0.2 | 52.3 ± 0.2 | 81.5 ± 0.5 | - |
| GNN-based methods | PATCHY-SAN | 73.1 ± 2.4 | - | 72.6 ± 2.2 | 49.1 ± 0.7 |
| | DGCNN | 70.0 ± 0.9 | 50.0 ± 1.9 | 73.7 ± 0.0 | - |
| | CapsGNN | 73.1 ± 4.8 | 51.1 ± 3.1 | 79.6 ± 2.9 | 52.8 ± 1.4 |
| | GIN* | 75.1 ± 5.1 | 52.3 ± 2.8 | 80.2 ± 1.9 | 57.0 ± 1.7 |
| Unsupervised methods | InfoGraph | 73.0 ± 0.9 | 49.7 ± 0.5 | 70.7 ± 1.1 | 53.4 ± 1.0 |
| | GraphCL | 71.1 ± 0.4 | 49.2 ± 0.6 | 71.4 ± 1.2 | 55.9 ± 0.2 |
| | MVGRL | 74.2 ± 0.7 | 51.2 ± 0.5 | 76.0 ± 1.2 | - |
| | AutoGCL | 73.3 ± 0.4 | - | 70.1 ± 0.7 | 56.7 ± 0.2 |
| | JOAO | 70.2 ± 3.0 | - | 69.5 ± 0.3 | 55.7 ± 0.6 |
| Our work | StructPosGSSL-SA | **75.5 ± 3.3** | **52.8 ± 3.5** | 76.5 ± 3.6 | **57.2 ± 3.3** |
| | StructPosGSSL-FA | 75.1 ± 3.1 | 52.3 ± 3.6 | 76.1 ± 3.3 | 56.8 ± 3.4 |

**Table 3: Classification accuracy (%) on social network datasets averaged over 10 runs.**

| Method | ogbg-molhiv | ogbg-moltox21 | ogbg-moltoxcast | ogbg-ppa | ogbg-molpcba |
|---|---|---|---|---|---|
| GIN [62] | 75.58 ± 1.40 | 74.91 ± 0.51 | 63.41 ± 0.74 | 68.92 ± 1.00 | 22.66 ± 0.28 |
| GIN+VN [24] | 75.20 ± 1.30 | 76.21 ± 0.82 | 66.18 ± 0.68 | 70.37 ± 1.07 | 27.03 ± 0.23 |
| GSN [7] | 77.99 ± 1.00 | - | - | - | - |
| ID-GNN [65] | 78.30 ± 2.00 | - | - | - | - |
| Deep LRP [11] | 77.19 ± 1.40 | - | - | - | - |
| GraphSNN [57] | 78.51 ± 1.70 | 75.45 ± 1.10 | 65.40 ± 0.71 | 70.66 ± 1.65 | 24.96 ± 1.50 |
| DS-GNN (EGO+) [5] | 77.40 ± 2.19 | 76.39 ± 1.18 | - | - | - |
| DSS-GNN (EGO+) [5] | 76.78 ± 1.66 | 77.95 ± 0.40 | - | - | - |
| POLICY-LEARN [4] | 78.49 ± 1.01 | 77.36 ± 0.60 | - | - | - |
| StructPosGSSL-SA | **78.80 ± 2.27** | **78.50 ± 2.30** | 66.50 ± 2.60 | **70.81 ± 1.45** | 27.10 ± 1.65 |
| StructPosGSSL-FA | **79.00 ± 2.43** | 77.60 ± 2.25 | 67.00 ± 2.20 | 70.90 ± 1.86 | 27.45 ± 1.95 |

**Table 4: Class. acc. (%) on OGB datasets averaged over 10 runs.**

## 6.3 Experiments on Synthetic Graphs

We use 2 publicly accessible datasets : (1) the Circular Skip Link (CSL) dataset [41]; and (2) SR25 [1]. Both benchmarks involve classifying graphs into isomorphism classes. CSL dataset, initially presented by [41] and frequently utilized to assess graph expressiveness [15], comprises 10 isomorphism classes of 41-node 4-regular graphs, almost all of which can be distinguished by the 3-WL test. SR25 dataset [1] comprises 15 strongly regular graphs, each consisting of 25 nodes, which cannot be distinguished by the 3-WL test.

We compare our approach against the five baselines: GCN [29], GIN [23], 3WLGNN [37], 3-GCN [40], and GCN-RNI [1]. We use the Adam optimizer with a learning rate of 0.001, a batch size of 32, dropout of 0.7, positional encoding dimension of 6, a temperature scaling parameter $\tau$ of 0.10, and run training for 500 epochs across both datasets. We use a 3-layer MLP with a hidden dimension of

| Method | CSL | SR25 |
|---|---|---|
| StructPosGSSL-SA [POS] | 54.7 ± 3.7 | 65.3 ± 3.5 |
| StructPosGSSL-FA [POS] | 82.7 ± 2.5 | 88.3 ± 2.8 |
| StructPosGSSL-SA [CW] | **92.0 ± 2.8** | **93.5 ± 3.7** |
| StructPosGSSL-FA [CW] | 91.3 ± 3.2 | 92.5 ± 2.8 |

**Table 5: Ablation study classification accuracy (%) for CSL and SR25 test sets.**

200 and a number of hops $k = 3$ for both settings. We choose $\lambda_{inv} = 1, \lambda_{var} = 25, \lambda_{cov} = 25$, and $\mu = 0.0009$ for the CSL dataset and $\lambda_{inv} = 1, \lambda_{var} = 24, \lambda_{cov} = 24$, and $\mu = 0.005$ for the SR25 dataset. In Table 1, we present the average and standard deviation obtained from 10-fold cross-validation.

Note that none of the baselines achieved the best performance on both synthetic datasets we evaluated, compared to the other baselines. To address **Q3**, as shown in Table 1, StructPosGSSL consistently achieves the best performance on both synthetic datasets. Specifically, *StructPosGSSL* improves upon the best results of the baselines by a margin of 0.8% (3WLGNN) and 93.4% (GCN, GIN, 3-GCN, and GCN-RNI) on the datasets CSL and SR25, respectively.

## 6.4 Ablation Analysis of Structural and Positional Encoding

To showcase the effectiveness of structural and positional information, we perform an ablation study on the following variants:

- **POS**: This variant excludes only Positional (POS) encoding.
- **CW**: This variant keeps only Closed-Walk (CW) information.

We performed an ablation study on the *StructPosGSSL* variants. The results presented in Table 5 demonstrate that the closed-walk structural information plays a key role in performance. Notably, this structural information has the most significant influence, while positional information has the least impact on the CSL and SR25 datasets, as shown in Table 5.

To address the **Q4**, performance on the CSL dataset decreases by 43.9% on StructPosGSSL-SA [POS], by 15.6% on StructPosGSSL-FA [POS], 6.6% on StructPosGSSL-SA [CW], and 7% on StructPosGSSL-FA [CW], compared to the original performance shown in Table 1. Similarly, performance on the SR25 dataset decreases by 34.7% on StructPosGSSL-SA [POS], by 11.7% on StructPosGSSL-FA [POS], 6.5% on StructPosGSSL-SA [CW], and 7.5% on StructPosGSSL-FA [CW], compared to the original performance shown in Table 1.

## 7 Conclusions

In conclusion, our proposed *StructPosGSSL* framework effectively addresses a key limitation in Graph Self-Supervised Learning by improving the capture of topological information. Leveraging the k-hop message-passing mechanism of *GenHopNet* and the integration of structural and positional awareness, *StructPosGSSL* exceeds the expressiveness of traditional GNNs and the Weisfeiler-Lehman test. Our experimental results show that the framework delivers superior performance on graph classification tasks, enhancing accuracy while maintaining computational efficiency. This advancement significantly strengthens GSSL's capability to distinguish between graphs with similar local structures but distinct global topologies.

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

# A   Appendix

## A.1   Laplacian Eigenvectors for Positional Encoding

Positional features should ideally differentiate nodes that are far apart in the graph while ensuring that nearby nodes have similar features. We use graph Laplacian eigenvectors as node positional features because they have fewer ambiguities and more accurately represent distances between nodes [15, 51]. Laplacian eigenvectors can embed graphs into Euclidean space, providing a meaningful local coordinate system while preserving the global graph structure. They are mathematically defined by the factorization of the graph Laplacian matrix as $\mathbf{L} = \mathbf{U}\Lambda\mathbf{U}^H$, where $\mathbf{U} = \{\mathbf{u}_i\}_{i=1}^m \in \mathbb{R}^m$ are orthogonal eigenvectors, $\Lambda = diag\left([\lambda_1, \ldots, \lambda_m]\right) \in \mathbb{R}^{m \times m}$ are real eigenvalues, and $\mathbf{U}^H$ is a hermitian transpose of $U$. After normalizing to unit length, eigenvectors are defined up to a factor of $\pm 1$, leading to random sign flips during training. In our experiments, we employ the $p$ smallest non-trivial eigenvectors, with $p$ specified for each experiment. The initial positional encoding vector for each node is computed beforehand and assigned as node attributes during dataset creation.

## A.2   Proofs of Lemmas and Theorems

**Theorem 1** *The following statement is true: (a) If $\sum_k \mathbf{A}_{vv}^k \simeq_{Cycle} \sum_k \mathbf{A}_{v'v'}^k$, then $\sum_k \mathbf{A}_{vv}^k \simeq_{ClosedWalk} \sum_k \mathbf{A}_{v'v'}^k$; but not vice versa.*

PROOF. The implication $\sum_k \mathbf{A}_{vv}^k \simeq_{Cycle} \sum_k \mathbf{A}_{v'v'}^k \Rightarrow \sum_k \mathbf{A}_{vv}^k \simeq_{ClosedWalk} \sum_k \mathbf{A}_{v'v'}^k$ is true because every cycle is a closed walk, but not every closed walk is a cycle. Thus, if two nodes are isomorphic with respect to cycles, they must also be isomorphic with respect to closed walks. The reverse implication $\sum_k \mathbf{A}_{vv}^k \simeq_{ClosedWalk} \sum_k \mathbf{A}_{v'v'}^k \Rightarrow \sum_k \mathbf{A}_{vv}^k \simeq_{Cycle} \sum_k \mathbf{A}_{v'v'}^k$ is not true because closed walks can include walks that repeat vertices or edges, which do not qualify as cycles. □

**Theorem 2** *Let $S$ represent a GNN with an aggregation scheme $\Phi$ delineated by Eq. 1-Eq. 4. $S$ exceeds the expressiveness of 1-WL in identifying non-isomorphic graphs, provided that $S$ operates over a sufficient number of hops, where $k > 1$, and also meets the following criteria:*

*(1)* $\Phi\left(\mathbf{h}_v^{(t)}, \{\!\!\{(\mathbf{A}_{vu}, \mathbf{h}_u^{(t)}, e_{uv}^b, e_{uv}^c) | u \in \mathcal{N}(v)\}\!\!\}, \{\!\!\{(\tilde{\mathbf{A}}_{vu}^k, \mathbf{h}_u^{(t)}) | u \in \mathcal{N}^k(v)\}\!\!\}, \{\!\!\{(\mathbf{A}_{vv}^k, \mathbf{h}_v^{(t)})\}\!\!\}\right)$ *is injective;*

*(2) The graph-level readout function of $S$ is injective.*

PROOF. For the proof, we proceed in two steps. First, we assume the existence of two graphs $G_1$ and $G_2$ that are distinguishable by 1-WL but indistinguishable by $S$, and we demonstrate a contradiction. We consider the iterations of 1-WL from 1 to $k$, where $k$ is the number of hops. If 1-WL distinguishes $G_1$ and $G_2$ using the information up to the $k$-th iteration but $S$ cannot, it implies the existence of k-hop local neighborhood subgraphs $\mathcal{G}_i$ and $\mathcal{G}_j$ with different multisets of $\mathbf{H} \in \mathcal{H}$, $\mathbf{W}_1 \in \mathcal{W}_1$, $\mathbf{W}_2 \in \mathcal{W}_2$, $\mathbf{W}_3 \in \mathcal{W}_3$. However, by the injectiveness property of $\Phi$, $S$ should yield different tuple $(\mathbf{H}, \mathbf{W}_1, \mathbf{W}_2, \mathbf{W}_3)$ for $\mathcal{G}_i$ and $\mathcal{G}_j$, contradicting the assumption. In the second step, we prove the existence of at least two graphs

| Variants | MUTAG | PTC-MR | NCI1 | PROTEINS | IMDB-B | IMDB-M |
|---|---|---|---|---|---|---|
| NT-Xent+NoVICReg | 88.5 ± 3.5 | 63.1 ± 3.0 | 78.3 ± 3.1 | 70.7 ± 3.9 | 72.0 ± 3.6 | 49.0 ± 3.6 |
| NT-Xent+Inv | 90.0 ± 3.4 | 64.3 ± 4.1 | 79.3 ± 3.2 | 71.7 ± 4.1 | 73.1 ± 4.1 | 49.8 ± 3.3 |
| NT-Xent+Var | 90.5 ± 3.6 | 64.8 ± 4.2 | 79.9 ± 3.6 | 72.8 ± 4.6 | 73.8 ± 4.3 | 50.3 ± 4.3 |
| NT-Xent+Cov | 92.0 ± 3.4 | 66.0 ± 3.2 | 81.0 ± 3.4 | 73.9 ± 4.3 | 74.2 ± 3.4 | 51.6 ± 4.1 |

**Table 6: Classification accuracy (%) averaged over 10 runs.**

distinguishable by $S$ but indistinguishable by 1-WL. This step involves providing specific examples of such graphs, illustrating $S$'s enhanced expressiveness compared to 1-WL. By completing these steps, we establish the validity of Theorem A.2, confirming that under the specified conditions, $S$ indeed surpasses the expressiveness of 1-WL in identifying non-isomorphic graphs. □

**Lemma 1** *Given two distinct pairs of multisets* $\mathbf{H}_1, \mathbf{H}_1' \in \mathcal{H}$, $\mathbf{W}_1, \mathbf{W}_1' \in \mathcal{W}_1$, $\mathbf{W}_2, \mathbf{W}_2' \in \mathcal{W}_2$, $\mathbf{W}_3, \mathbf{W}_3' \in \mathcal{W}_3$, *there exists a function* $f$ *such that the aggregation function* $\pi(\mathbf{h}_v, \mathbf{H}_1, \mathbf{W}_1, \mathbf{W}_2, \mathbf{W}_3)$ *and* $\pi(\mathbf{h}_v', \mathbf{H}_1', \mathbf{W}_1', \mathbf{W}_2', \mathbf{W}_3')$ *defined as* $\pi(\mathbf{h}_v, \mathbf{H}_1, \mathbf{W}_1, \mathbf{W}_2, \mathbf{W}_3) = \sum_{\mathbf{h} \in \mathbf{H}_1, \mathbf{w}_1 \in \mathbf{W}_1} f(\mathbf{h}, \mathbf{w}_1) + \sum_k \left( f(\mathbf{h}_v, \mathbf{w}_3) + \sum_{\mathbf{h} \in \mathbf{H}_1, \mathbf{w}_2 \in \mathbf{W}_2} f(\mathbf{h}, \mathbf{w}_2) \right)$ *and* $\pi(\mathbf{h}_v', \mathbf{H}_1', \mathbf{W}_1', \mathbf{W}_2', \mathbf{W}_3') = \sum_{\mathbf{h} \in \mathbf{H}_1', \mathbf{w}_1 \in \mathbf{W}_1'} f(\mathbf{h}, \mathbf{w}_1) + \sum_k \left( f(\mathbf{h}_v', \mathbf{w}_3) + \sum_{\mathbf{h} \in \mathbf{H}_1', \mathbf{w}_2 \in \mathbf{W}_2'} f(\mathbf{h}, \mathbf{w}_2) \right)$ *are unique, respectively.*

PROOF. Since $\mathcal{H}, \mathcal{W}_1, \mathcal{W}_2$, and $\mathcal{W}_3$ are countable, there must exist four functions $\psi_1 : \mathcal{H} \to \mathbb{N}_{odd}$ mapping $\mathbf{h} \in \mathcal{H}$ to odd natural numbers and $\psi_2, \psi_3$, and $\psi_4$ mapping elements from $\mathcal{W}_1, \mathcal{W}_2$, and $\mathcal{W}_3$ to even natural numbers, respectively. For any pair of multisets $(\mathbf{h}_v, \mathbf{H}_1, \mathbf{W}_1, \mathbf{W}_2, \mathbf{W}_3)$, given that the cardinalities of $\mathbf{H}_1, \mathbf{W}_1, \mathbf{W}_2$, and $\mathbf{W}_3$ are bounded, there must be a natural number $N$ such that $|\mathbf{H}_1| < N$, $|\mathbf{W}_1| < N$, $|\mathbf{W}_2| < N$, and $|\mathbf{W}_3| < N$. Consider a prime number $P > 4N$. Define the function $f$ such that $f(\mathbf{h}, \mathbf{w}_1, \mathbf{w}_2, \mathbf{w}_3) = P^{-\psi_1(\mathbf{h})} + P^{-\psi_2(\mathbf{w}_1)} + P^{-\psi_3(\mathbf{w}_2)} + P^{-\psi_4(\mathbf{w}_3)}$. Then, the aggregation functions $\pi(\mathbf{h}_v, \mathbf{H}_1, \mathbf{W}_1, \mathbf{W}_2, \mathbf{W}_3)$ and $\pi(\mathbf{h}_v', \mathbf{H}_1', \mathbf{W}_1', \mathbf{W}_2', \mathbf{W}_3')$ are unique for each distinct pair of multisets because the sum of these functions will be unique for distinct pairs of multisets by the properties of prime numbers and the unique mappings $\psi_1, \psi_2, \psi_3$, and $\psi_4$. □

**Lemma 2** *Expanding upon Lemma 1, we introduce an extended aggregation function* $\pi(\mathbf{h}_v, \mathbf{H}, \mathbf{W}_1, \mathbf{W}_2, \mathbf{W}_3)$, *which incorporates the feature vector of the central node* $h_v$ *and the multisets* $\mathbf{H} \in \mathcal{H}$, $\mathbf{W}_1 \in \mathcal{W}_1$, $\mathbf{W}_2 \in \mathcal{W}_2$, *and* $\mathbf{W}_3 \in \mathcal{W}_3$. *There exists a function* $f$ *such that* $\pi(\mathbf{h}_v, \mathbf{H}, \mathbf{W}_1, \mathbf{W}_2, \mathbf{W}_3) = (1 + \epsilon)f(\mathbf{h}_v) + \sum_{\mathbf{h} \in \mathbf{H}, \mathbf{w}_1 \in \mathbf{W}_1} f(\mathbf{h}, \mathbf{w}_1) + \sum_k \left( f(\mathbf{h}_v, \mathbf{w}_3) + \sum_{\mathbf{h} \in \mathbf{H}, \mathbf{w}_2 \in \mathbf{W}_2} f(\mathbf{h}, \mathbf{w}_2) \right)$ *is unique for any distinct quintuples* $(\mathbf{h}_v, \mathbf{H}, \mathbf{W}_1, \mathbf{W}_2, \mathbf{W}_3)$, *where* $\mathbf{h}_v \in \mathcal{H}$, $\mathbf{w}_3 \in \mathcal{W}_3$, *and* $\epsilon$ *is an arbitrary real number.*

PROOF. Let $(\mathbf{h}_v, \mathbf{H}_1, \mathbf{W}_1, \mathbf{W}_2, \mathbf{W}_3)$ and $(\mathbf{h}_v', \mathbf{H}_1', \mathbf{W}_1', \mathbf{W}_2', \mathbf{W}_3')$ be two different tuples. Then, there are two cases:

(1) When $\mathbf{h}_v = \mathbf{h}_v'$ but $(\mathbf{h}_v, \mathbf{H}_1, \mathbf{W}_1, \mathbf{W}_2, \mathbf{W}_3) \neq (\mathbf{h}_v', \mathbf{H}_1', \mathbf{W}_1', \mathbf{W}_2', \mathbf{W}_3')$, by Lemma A.2, we know that $\sum_{\mathbf{h} \in \mathbf{H}_1, \mathbf{w}_1 \in \mathbf{W}_1} f(\mathbf{h}, \mathbf{w}_1) + \sum_k \left( f(\mathbf{h}_v, \mathbf{w}_3) + \sum_{\mathbf{h} \in \mathbf{H}_1, \mathbf{w}_2 \in \mathbf{W}_2} f(\mathbf{h}, \mathbf{w}_2) \right) \neq$

$\sum_{\mathbf{h} \in \mathbf{H}_1', \mathbf{w}_1 \in \mathbf{W}_1'} f(\mathbf{h}, \mathbf{w}_1) + \sum_k \left( f(\mathbf{h}_v', \mathbf{w}_3) + \sum_{\mathbf{h} \in \mathbf{H}_1', \mathbf{w}_2 \in \mathbf{W}_2'} f(\mathbf{h}, \mathbf{w}_2) \right)$. Thus, $\pi(\mathbf{h}_v, \mathbf{H}_1, \mathbf{W}_1, \mathbf{W}_2, \mathbf{W}_3) \neq \pi(\mathbf{h}_v', \mathbf{H}_1', \mathbf{W}_1', \mathbf{W}_2', \mathbf{W}_3')$.

(2) When $\mathbf{h}_v \neq \mathbf{h}_v'$, we prove $\pi(\mathbf{h}_v, \mathbf{H}_1, \mathbf{W}_1, \mathbf{W}_2, \mathbf{W}_3) \neq \pi(\mathbf{h}_v', \mathbf{H}_1', \mathbf{W}_1', \mathbf{W}_2', \mathbf{W}_3')$ by contradiction. Assume that $\pi(\mathbf{h}_v, \mathbf{H}_1, \mathbf{W}_1, \mathbf{W}_2, \mathbf{W}_3) = \pi(\mathbf{h}_v', \mathbf{H}_1', \mathbf{W}_1', \mathbf{W}_2', \mathbf{W}_3')$. Then, we have:

$$(1 + \epsilon)f(\mathbf{h}_v) + \sum_{\mathbf{h} \in \mathbf{H}, \mathbf{w}_1 \in \mathbf{W}_1} f(\mathbf{h}, \mathbf{w}_1) +$$
$$\sum_k \left( f(\mathbf{h}_v, \mathbf{w}_3) + \sum_{\mathbf{h} \in \mathbf{H}, \mathbf{w}_2 \in \mathbf{W}_2} f(\mathbf{h}, \mathbf{w}_2) \right) =$$
$$(1 + \epsilon)f(\mathbf{h}_v') + \sum_{\mathbf{h} \in \mathbf{H}_1', \mathbf{w}_1 \in \mathbf{W}_1'} f(\mathbf{h}, \mathbf{w}_1) +$$
$$\sum_k \left( f(\mathbf{h}_v', \mathbf{w}_3) + \sum_{\mathbf{h} \in \mathbf{H}_1', \mathbf{w}_2 \in \mathbf{W}_2'} f(\mathbf{h}, \mathbf{w}_2) \right).$$

This gives us the following equation:

$$(1 + \epsilon)\left( f(\mathbf{h}_v) - f(\mathbf{h}_v') \right) = \sum_{\mathbf{h} \in \mathbf{H}, \mathbf{w}_1 \in \mathbf{W}_1} f(\mathbf{h}, \mathbf{w}_1) - \sum_{\mathbf{h} \in \mathbf{H}_1', \mathbf{w}_1 \in \mathbf{W}_1'} f(\mathbf{h}, \mathbf{w}_1)$$
$$+ \left( \sum_k \left( f(\mathbf{h}_v, \mathbf{w}_3) + \sum_{\mathbf{h} \in \mathbf{H}, \mathbf{w}_2 \in \mathbf{W}_2} f(\mathbf{h}, \mathbf{w}_2) \right) \right)$$
$$- \left( \sum_k \left( f(\mathbf{h}_v', \mathbf{w}_3) + \sum_{\mathbf{h} \in \mathbf{H}_1', \mathbf{w}_2 \in \mathbf{W}_2'} f(\mathbf{h}, \mathbf{w}_2) \right) \right).$$

If $\epsilon$ is an irrational number, the left-hand side of the equation is irrational, while the right-hand side is rational, leading to a contradiction. Therefore, $\pi(\mathbf{h}_v, \mathbf{H}_1, \mathbf{W}_1, \mathbf{W}_2, \mathbf{W}_3) \neq \pi(\mathbf{h}_v', \mathbf{H}_1', \mathbf{W}_1', \mathbf{W}_2', \mathbf{W}_3')$.

□

**Corollary 1** *GenHopNet exhibits greater expressiveness compared to 1-WL when evaluating non-isomorphic graphs.*

PROOF. We demonstrate this theorem by proving that *GenHopNet* is a GNN that meets the conditions specified in Theorem 2. For the first condition, consider the two graphs depicted in Figure 1(b). *GenHopNet* can differentiate these graphs as $\{\!\{ (A_{vu}, h_u^{(t)}, e_{uv}^b, e_{uv}^c) | u \in \mathcal{N}(v) \}\!\} \neq \{\!\{ (A_{v'u'}, h_{u'}^{(t)}, e_{u'v'}^b, e_{u'v'}^c) | u' \in \mathcal{N}(v') \}\!\}$, $\{\!\{ (\tilde{A}_{ou}^k, h_u^{(t)}) | u \in \mathcal{N}^k(v) \}\!\} \neq \{\!\{ (\tilde{A}_{v'u'}^k, h_{u'}^{(t)}) | u' \in \mathcal{N}^k(v') \}\!\}$, and $\{\!\{ (A_{vv}^k, \mathbf{h}_v^{(t)}) \}\!\} \neq \{\!\{ (A_{v'v'}^k, \mathbf{h}_{v'}^{(t)}) \}\!\}$. For the second condition, leveraging Lemmas 1 and 2, along with the fact that an MLP can serve as a universal approximator [62] to model and learn the functions $f$ and $g$, we establish that *GenHopNet* also satisfies this condition. □

**Theorem 3** StructPosGSSL *is more expressive than subgraph MPNNs in distinguishing certain non-isomorphic graphs.*

Proof. To prove Theorem 3, we consider the two non-isomorphic graphs $G_1$ and $G_2$ shown in Figure 4. Let $v \in G_1$ and $v' \in G_2$ be the middle nodes in each graph. In the case of Subgraph MPNNs (e.g., [12, 65, 71]), the aggregation function over the neighborhoods of $v$ and $v'$ fails to differentiate between the two nodes. This is because Subgraph MPNNs rely on local subgraphs, and the structural features and neighborhood-based information are symmetric for $v$ and $v'$.

Let us first consider the structural encoder (i.e., *GenHopNet*) with only closed-walk information up to $k = 3$, without EB attributes $e_{uv}^c$. In this case, the node representations for $v$ and $v'$ generated by the structural encoder using closed-walks are identical, since $\{\!\{(A_{vu}, h_u^{(t)}, e_{uv}^b)|u \in \mathcal{N}(v)\}\!\} = \{\!\{(A_{v'u'}, h_{u'}^{(t)}, e_{u'v'}^b)|u' \in \mathcal{N}(v')\}\!\}$, $\{\!\{(\tilde{A}_{vu}^k, h_u^{(t)})|u \in \mathcal{N}^k(v)\}\!\} = \{\!\{(\tilde{A}_{v'u'}^k, h_{u'}^{(t)})|u' \in \mathcal{N}^k(v')\}\!\}$, and $\{\!\{(A_{vv}^k, \mathbf{h}_v^{(t)})\}\!\} = \{\!\{(A_{v'v'}^k, \mathbf{h}_{v'}^{(t)})\}\!\}$. This indicates that the closed-walk information alone is insufficient to distinguish these non-isomorphic graphs. Now, we show how *StructPosGSSL*, when enhanced with positional encodings and $e_{uv}^c$, can differentiate between the non-isomorphic graphs $G_1$ and $G_2$. Let $h_{u,pos}^{(t)}$ be the positional encoding of node $u$. When positional encodings are combined with $e_{uv}^c$, the aggregation function can distinguish these non-isomorphic graph pairs. Using Lemmas 1 and 2, we know that the aggregation function is still injective when positional encodings and $e_{uv}^c$ are included. Thus, for the middle nodes of each graph, we have $\{\!\{(A_{vu}, h_{u,pos}^{(t)}, e_{uv}^b, e_{uv}^c)|u \in \mathcal{N}(v)\}\!\} \neq \{\!\{(A_{v'u'}, h_{u',pos}^{(t)}, e_{u'v'}^b, e_{u'v'}^c)|u' \in \mathcal{N}(v')\}\!\}$. Therefore, the *StructPosGSSL* with positional encodings and EB attributes yields different representations for $v$ and $v'$, even though they were previously indistinguishable. □

## A.3 Ablation Analysis of Loss Function

To showcase the effectiveness of each element in the loss function, we perform an ablation study on the following variants:

- NoVICReg: This variant excludes the VICReg regularization term from the overall loss.
- Inv: This variant keeps only the Invariance term in the VICReg regularization term.
- Var: This variant keeps only the Variance term in the VICReg regularization term.
- Cov: This variant keeps only the Covariance term in the VICReg regularization term.

We conducted the ablation study on the *StructPosGSSL-SA* variant. The results shown in Table 6 indicate that the Invariance, Variance, and Covariance terms are crucial to the performance. Specifically, the covariance term has the greatest impact on performance, whereas the invariance term has the least effect across all datasets, as detailed in Table 6. Specifically, as shown in Table 6, performance on graphs decreases by 3.5% to 5.1% with the NT-Xent+NoVICReg loss function, by 2.4% to 4.1% with NT-Xent+Inv, by 1.7% to 3.0% with NT-Xent+Var, and by 1.0% to 1.9% with NT-Xent+Cov, compared to the combined NT-Xent+VICReg loss function.

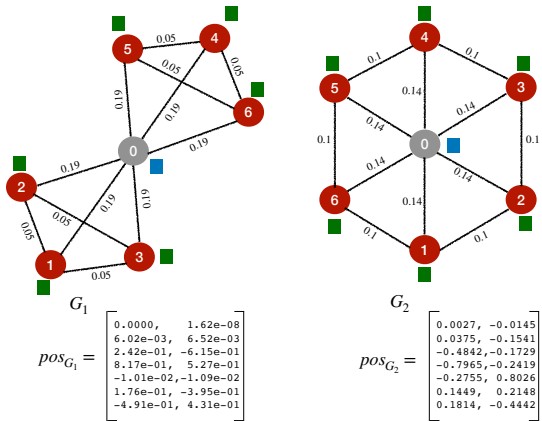

$$pos_{G_1} = \begin{bmatrix} 0.0000, & 1.62e\text{-}08 \\ 6.02e\text{-}03, & 6.52e\text{-}03 \\ 2.42e\text{-}01, & -6.15e\text{-}01 \\ 8.17e\text{-}01, & 5.27e\text{-}01 \\ -1.01e\text{-}02, -1.09e\text{-}02 \\ 1.76e\text{-}01, -3.95e\text{-}01 \\ -4.91e\text{-}01, & 4.31e\text{-}01 \end{bmatrix}$$

$$pos_{G_2} = \begin{bmatrix} 0.0027, & -0.0145 \\ 0.0375, & -0.1541 \\ -0.4842, & -0.1729 \\ -0.7965, & -0.2419 \\ -0.2755, & 0.8026 \\ 0.1449, & 0.2148 \\ 0.1814, & -0.4442 \end{bmatrix}$$

**Figure 4: A pair of non-isomorphic graphs where the colored square box on each node represents the feature representations derived from closed-walk information. The two middle nodes (colored gray) in graphs $G_1$ and $G_2$ cannot be distinguished using only closed-walk information (up to $k = 3$), as they receive the same representation (colored blue). However, by incorporating positional information along with EB attributes, we can successfully distinguish these nodes.**

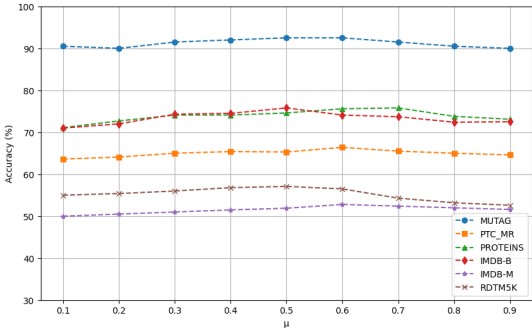

**Figure 5: Accuracy (%) of StructPosGSSL-SA under different $\mu$ values.**

## A.4 Comparison under Different $\mu$

To evaluate the impact of the regularization term $\mu$ on the performance of our *StructPosGSSL* framework, we conduct experiments by evaluating *StructPosGSSL-SA* across six datasets (MUTAG, PTC-MR, PROTEINS, IMDB-B, IMDB-M, and RDT5K), using varying values for the regularization term $\mu = [0.1, 0.2, \ldots, 0.9]$. For this experimental setup, we use the same hyperparameter configuration for each dataset as described in Section 6.1. Figure 5 represents the experimental results. In our experiments, we observed that setting $\mu$ either too low or too high leads to suboptimal performance. To achieve better results, it is essential to select an intermediate value for $\mu$, as this provides a balance that optimizes the *StructPosGSSL*'s performance.

