# OpenReview forum: "Graph Self-Supervised Learning with Learnable Structural and Positional Encodings"
_ACM.org/TheWebConf/2025/Conference — WWW 2025 Poster_

### Official Review · Reviewer_VJTo · 2024-11-10

**Novelty:** 4
**Technical Quality:** 4

**Review:**

This paper presents a well-structured framework (StructPosGSSL) that effectively addresses limitations in existing Graph Self-Supervised Learning methods.
Pros:
1. The motivation is clear. In particular, the integration of structural and positional information is easy to understand.
2. The authors conducts lots of experiments.
3.The writing is generally clear and concise, with a logical flow that guides the reader through the problem statement, methodology, and results.
Cons:
1. The idea is not very novel to me and combines some existing components.
2. The experimental results are not impressive to me. In fact, the proposed method even performs worse in many cases.
3. Most of the comparison methods are quite old. The authors fail to compare with transformer-based methods.

**Questions:**

Why the proposed method fails to beat other existing methods?

**Reviewer Confidence:**

3: The reviewer is confident but not certain that the evaluation is correct

**Scope:**

3: The work is somewhat relevant to the Web and to the track, and is of narrow interest to a sub-community

---

### Official Review · Reviewer_PJCs · 2024-11-11

**Novelty:** 4
**Technical Quality:** 3

**Review:**

The paper introduce a structural- and positional-aware GSSL framework, StructPosGSSL, enhancing the global topology understanding and the sensitivity to specific structural and feature augmentations. Experiment results demonstrate that StructPosGSSL significantly advances GSSL’s capability in distinguishing graphs with similar local structures but different global topologies.

**Pros:**
1. The idea of introducing structural and positional module is reasonable. And the whole framework is based on contrastive learning, making it more robust and carefully designed.

2. The paper carefully and theoretically demonstrate that GenHopNet surpasses the expressiveness of the classical WL test for graph isomorphism.

*Cons:*
1. The baseline methods used in the experiment section are too old (before 2022). The paper does not include those newest SOTA methods. And the ablation analysis section does not prove that all terms (like 1-hop edge-level aggregated message) in Eq.(5) are necessary.

**Questions:**

1.  In Section 5.2, it seems that the graph encoder only considers only eigenvalues and edge attributes. How do you combine these terms exactly?

2. In the ablation study, the author does not offer the original StructPosGSSL performance. I can't tell which module is effective from the table. It seems that the positional module is much more effective that the structural one, why?

Please answer the above questions and cons, I will consider raising the score.

**Reviewer Confidence:**

2: The reviewer is willing to defend the evaluation, but it is likely that the reviewer did not understand parts of the paper

**Scope:**

3: The work is somewhat relevant to the Web and to the track, and is of narrow interest to a sub-community

---

### Official Review · Reviewer_RwQK · 2024-11-30

**Novelty:** 5
**Technical Quality:** 5

**Review:**

This paper proposes a novel GNN model and a new graph self-supervised learning paradigm, aiming at problems in graph classification and graph self-supervised learning. The proposed GenHopNet defines the new aggregation scheme to incorporate further neighborhood information and distinguish small structural differences. It theoretically shows GenHopNet representations' uniqueness so that the model can distinguish certain non-isomorphic graphs. The authors conduct extensive experiments to show the effectiveness of the proposed methods.
However, I think there are some weaknesses.

Cons:
1. The article is not very well organized. Although the title implies more on Graph SSL, the article focuses more on the new GNN architecture. The proposed SSL framework seems to contribute much less than the proposed GenHopNet.
2. I think that Def. 1 and 2 are redundant and their role in benefiting GenHopNet is not reflected.
3. It gives the impression, that the proposed new SSL method, is a combination of two contrastive learning methods.

**Questions:**

1. For Lemma. 1 and 2, do they still hold if we exchange the aggregation function $\pi$ with the common one in GCN? i.e., $\pi(h_v, H_1, W_1)= (1+\epsilon)f(h_v) + \sum_{h \in H_1, w_1 \in W_1}f(h, w_1)$.
2. If we only use GenHopNet with the traditional SSL paradigm, i.e., using NT-Xent loss after getting graph representations from GenHopNet, will it still outperform other SSL methods using traditional GNN backbones like GCNs?
3. How does the proposed model benefit from contrastive learning between Structural and Positional Encoding?
4. The experimental results show that the proposed SSL framework works. But which of the two loss functions is more important?

**Reviewer Confidence:**

3: The reviewer is confident but not certain that the evaluation is correct

**Scope:**

4: The work is relevant to the Web and to the track, and is of broad interest to the community

---

### Official Review · Reviewer_qs4w · 2024-12-03

**Novelty:** 5
**Technical Quality:** 4

**Review:**

> Summary:

This paper incorporates structural and positional embeddings (SE & PE) into graph self-supervised learning to improve the expressiveness of the learned graph representations. To this end, this paper proposes GenHopNet, a GNN with a k-hop message-passing mechanism, which breaks the limitation of the 1-WL test. Experimental results validate the effectiveness of the proposed method.

> Pros:

1. This paper is the first to study the expressiveness ability of GNNs in self-supervision learning.

2. The proposed method exhibits competitive performance in both real and synthetic datasets.

> Cons:

1. The expressiveness of GNNs has been well studied in supervision learning. Applying PE and SE to self-supervised learning is straightforward and somewhat incremental.

2. The theoretical analysis shows that GenHopNet can break the 1-WL test. However, I wonder whether the expressive power of GenHopNet is related to the self-supervised learning objectives or is simply due to the existence of PE & SE.

3. The datasets are too small to validate the effectiveness of the proposed method. It would be better to conduct experiments on the large-scale MoleculeNet [1].

[1] MoleculeNet: A Benchmark for Molecular Machine Learning.

**Questions:**

See weaknesses.

**Reviewer Confidence:**

3: The reviewer is confident but not certain that the evaluation is correct

**Scope:**

3: The work is somewhat relevant to the Web and to the track, and is of narrow interest to a sub-community